Characteristics of the retinal and choroidal thicknesses in myopic young adult males using swept-source optical coherence tomography

Alrasheed Saif Hassan 1 S.rasheed@qu.edu.sa
http://orcid.org/0000-0001-9979-4450 Gammoh Yazan 2
1 Department of Optometry, College of Applied Medical Sciences, Qassim University , Buraydah , Saudi Arabia
2 Department of Optometry Science, Faculty of Allied Medical Sciences, Al-Ahliyya Amman University , Amman , Jordan
Redondo Beatriz
Electronic publication date: 2025 Mar 5
Publication date: 2025
Volume: 13
Electronic Location ID: e19030
Received 2024 Aug 23; Accepted 2025 Jan 29
Copyright: © 2025 Alrasheed and Gammoh
Copyright year: 2025
Copyright holder: Alrasheed and Gammoh
License: This is an open access article distributed under the terms of the Creative Commons Attribution License, which permits unrestricted use, distribution, reproduction and adaptation in any medium and for any purpose provided that it is properly attributed. For attribution, the original author(s), title, publication source (PeerJ) and either DOI or URL of the article must be cited.
License URL: https://creativecommons.org/licenses/by/4.0/

Keywords: Choroidal change, Myopia, Retina, Clinical feature, Young adult

Funding: The authors received no funding for this work.

==============================
Background

Changes in retinal and choroidal structures are key biomarkers for predicting, diagnosing, and monitoring various ocular conditions, including myopia.

Objective

To assess the characteristics of the retinal and choroidal thicknesses in myopic young adult males using swept-source optical coherence tomography (SSOCT).

Methods

This cross-sectional comparative study included 198 young adults with a mean age of 21.87 ± 1.69 years, only male subjects were recruited for this study, comprising 102 diagnosed with myopia and 96 with emmetropia. Refraction was assessed using an autorefractometer, and comprehensive SSOCT scans were conducted to measure the thickness of the retina and choroid at nine predefined locations. Data analysis focused on identifying significant patterns and correlations between myopia and retinal and choroidal thickness.

Results

Myopic subjects with a mean of −2.66 ± 1.59D exhibited significantly decreased retinal thickness compared to emmetropic with a mean of 0.18 ± 0.39D, (p < 0.01). Similarly, their choroidal thickness was also significantly thinner (p < 0.01). The findings showed a weak but statistically significant inverse correlation between retinal thickness and the spherical equivalent of myopia (r = −0.257, p < 0.01). Correspondingly, a stronger inverse correlation was observed between choroidal thickness and the spherical equivalent of myopia (r = −0.306, p < 0.01). Choroidal thickness in all studied areas showed an inverse correlation with the degree of myopia (p < 0.05), except in the superior outer region, where the association was not statistically significant (p = 0.056).

Conclusion

The study identified significant differences in the retinal and choroidal structures between myopic and emmetropic individuals. The use of SSOCT effectively detected these morphological changes in myopic young adults, offering valuable insights into myopia’s pathophysiology and potentially guiding targeted therapeutic strategies for myopia control.

Introduction

The choroid, an extremely vascularized ocular structure, plays crucial roles in nourishing the retina and optic nerve, absorbing light, and regulating intraocular pressure and ocular growth (Xie et al., 2021). The blood-retina barrier is essential for maintaining an optimal retinal function. The inner barrier consists of tight junctions between retinal capillary endothelial cells. The outer barrier, formed by tight junctions between retinal pigment epithelial cells, regulates the movement of solutes and nutrients from the choroid to the sub-retinal space (Naylor et al., 2019). Choroidal thickness (ChT) is a key biomarker for predicting, diagnosing, intervening, and monitoring various retinal and choroidal diseases, including myopia (Prousali et al., 2021). The retina is an extension of the central nervous system. Thickness measurements of the macular retinal nerve fibre layer and ganglion cell-inner plexiform layer are commonly used as markers for early detection and follow-up of the ocular disorders (Banghart et al., 2022).

Optical coherence tomography (OCT) is a non-invasive imaging modality that provides high-resolution cross-sectional images of the retina and choroid. It allows for detailed assessment of retinal and choroid thickness, which is crucial for understanding retinal and choroidal diseases (Borrelli et al., 2018). Since OCT innovation in the 1990s, the device has significantly advanced clinical diagnosis and research compared to traditional methods like ultrasound and indocyanine green angiography (Branchini et al., 2012). The development of OCT-angiography has expanded OCT’s capabilities from structural to vascular imaging, enabling detailed quantitative analysis of ocular structures and vasculature, particularly retina and choroid (Xie et al., 2021; Branchini et al., 2012). Recent advancements in OCT technology have facilitated a deeper understanding of the morphological changes associated with myopia (Zhang et al., 2024). Swept-source optical coherence tomography (SSOCT) uses a long-wavelength light source, offering higher resolution, deeper and broader imaging, and faster scans than spectral-domain OCT (SDOCT). It is effective for epidemiologic studies of the fundus with consistent repeatability (Matsuo et al., 2013; Uzun & Pehlivan, 2016).

Myopia, commonly known as near-sightedness, is a predominant refractive error (RE) that affects a significant portion of the global population, particularly young adults (Alrasheed & Alghamdi, 2024). The recent increase in myopia prevalence, especially in urbanized regions, has raised concerns about its impact on visual health and quality of life (Holden et al., 2016; Alrasheed, Naidoo & Clarke-Farr, 2016). Myopia is usually associated with an increased risk of developing sight-threatening conditions such as retinal detachment, myopic maculopathy, and glaucoma, and is considered one of the leading causes of visual impairment, particularly in young adults (Flitcroft, 2012; Abdi Ahmed, Alrasheed & Alghamdi, 2020). Previous studies (Aydemir et al., 2021; Howaidy & Eldaly, 2021; Cheng & Tian, 2024) have demonstrated that myopic eyes often exhibit significant alterations in retinal and choroidal thickness. These changes include a thinner choroid, particularly in the macular region, and varying patterns of retinal layer thickness, which may be indicative of the mechanical stretching and elongation of the eye associated with myopia (Read et al., 2013). Understanding these structural changes is crucial for early detection and management of myopia-related complications.

Recent studies (Howaidy & Eldaly, 2021; Cheng & Tian, 2024; Alrasheed & Aldakhil, 2022) suggest that the choroid plays a role in ocular growth regulation and emmetropization, as refractive errors have been found to correlate with changes in ChT. The advent of SSOCT imaging has made it possible to qualitatively and quantitatively assess retinal and choroidal structures. Despite the growing body of literature, there remains a need for comprehensive studies focusing specifically on young adults, as this demographic is at a critical stage where myopia progression is often most pronounced (Alrasheed, Naidoo & Clarke-Farr, 2016; Flitcroft, 2012). Therefore, the present study aims to investigate the structural changes in retina and choroid in myopic young adults using SSOCT, providing insights that could contribute to improved clinical management and intervention strategies for myopia control.

Materials and Methods

Study design

This cross-sectional comparative hospital-based study was conducted on 198 young adult males aged 18 to 25 years, only male subjects were recruited for this study, with 102 diagnosed with myopia and 96 with emmetropia, at Qassim University’s optometry clinics from May to July 2024.

Sample size

This study used a non-probability sampling technique, with participants selected from Qassim University’s optometry clinic. The study included a sample size of 198 participants aged 18 to 25 years, comprising 102 diagnosed with myopia and 96 with emmetropia.

Inclusion criteria

The inclusion criteria for the study groups were adults diagnosed with myopia or emmetropia who agreed to participate and signed the consent form. Myopia was defined in present study as a spherical equivalent of refraction (SER) (sum of sphere + 1/2 cyl) from −1.00 to −7.00 D, and emmetropia as a SER between +0.75 D and −0.5 D. Exclusion criteria included presence of astigmatism more than −0.5 D, amblyopia, strabismus, ocular pathology that may had an effect on retinal imaging, history of corneal trauma, corneal refractive surgery, and systemic diseases.

Ethical approval

Ethical permission was obtained from the Qassim University Health Research Ethics Committee (approval number 21-10-03), and the study adhered to the Declaration of Helsinki guidelines. Written informed consent was obtained from all participants after explaining the study’s aim. Participants were able to withdraw from the study at any time, and no remuneration was offered for participation. Data were collected confidentially without any individual information being recorded.

Data collection procedures

Each participant underwent a history-taking process and comprehensive eye examinations, including measurement of visual acuity (VA) using a Snellen chart at a distance of 6 m and refraction assessments, to determine their refractive condition. Non-cycloplegic refraction was assessed using an autorefractometer (RK 4800; Topcon Corporation, Tokyo, Japan). Slit-lamp biomicroscopy was used to perform an anterior eye examination. For the posterior segment, a direct ophthalmoscope and 90D fundus biomicroscopy were utilized. Intraocular pressure was measured using non-contact tonometer to ensure participants met the study’s inclusion criteria. Comprehensive SSOCT scans were conducted to measure the thickness of the retina and choroid at nine predefined locations. SSOCT was performed using the Deep Range Imaging (DRI)-Triton SS-OCT Plus (Topcon Corporation, 75-1; Hasunuma-cho, Itabashi-ku, Tokyo, Japan). The device operates at a scanning speed of 100,000 A-scans per second and utilizes a wavelength of 1,050 nm. This allows for deeper tissue penetration, enabling detailed visualization of ocular structures such as the retina, choroid, and even the sclera. Retinal and choroidal thicknesses were automatically segmented and subsequently reviewed. Any segmentation errors were manually corrected, retested, or discarded. Image quality was assessed using a scale ranging from 0 to 100, with a minimum acceptable score of 60. Thickness measurements, expressed in micrometers (μm), were obtained for the nine regions of the Early Treatment Diabetic Retinopathy Study (ETDRS) grid using the 3D Macula protocol. The central area was 1 mm in diameter; the outer or perifoveal ring was 6 mm in diameter, divided into four areas: superior outer, temporal outer, nasal outer, and inferior outer; and the inner or parafoveal ring was 3 mm in diameter, divided into four areas: superior inner, temporal inner, nasal inner, and inferior inner, as shown in Fig. 1.

Figure 1 Retinal and choroidal thickness measurements obtained using the ETDRS grid with the 3D Macula protocol.

Data analysis

The data were entered into Microsoft Excel 2016 and reviewed by the principal investigator before being imported into Statistical Package for the Social Sciences (SPSS) software, version 25.0 (SPSS, Inc., Chicago, IL, USA). Descriptive statistics, including means and standard deviations, were used to summarize the data. The Kolmogorov-Smirnov (K-S) test was employed to assess data normality. The independent t-test test was used to compare the mean differences in retinal and choroidal thickness between myopic and emmetropic subjects. Data analysis focused on identifying significant patterns and correlations between the degree of myopia and the retina and choroid thickness. Pearson correlation analysis was focused on myopes because this group is more likely to exhibit variations in retinal and choroidal thickness due to their refractive status. Additionally, it aimed to enhance the understanding of the relationship between retinal and choroidal thickness in individuals with myopia. A p-value of less than 0.05 was considered indicative of statistical significance.

Results

Demographic characteristics

A total of 198 participants, aged 18 to 25 years, agreed to participate in the present study. The study sample included 102 young adult males with a mean age of 21.87 ± 1.69 years, diagnosed with myopia, and 96 young adult males with a mean age of 21.09 ± 1.75 years, diagnosed with emmetropia. The difference in age between the myopic and emmetropic subjects was not statistically significant (p = 0.235). The mean spherical equivalent for myopic eyes was −2.66 ± 1.59D, while for emmetropic eyes it was 0.18 ± 0.39D. The difference in the mean spherical equivalent between the myopic and emmetropic eyes was highly statistically significant (p = 0.001). The one-sample Kolmogorov-Smirnov test indicated that the measurements of retinal and choroidal thickness were normally distributed, p > 0.05.

Retinal and choroidal structures in myopic and emmetropic subjects

Myopic subjects exhibited a decrease in mean retinal thickness compared to emmetropic individuals, with the difference statistically significant (p < 0.01). Similarly, myopic subjects showed significantly thinner mean choroidal thickness than emmetropic subjects, with the difference also being statistically significant (p < 0.01). The central choroidal thickness was significantly thinner in myopic subjects compared to emmetropic subjects, with a statistically significant difference (p < 0.05). The superior inner choroidal thickness in myopic subjects was slightly thinner than in emmetropic subjects, although the difference was not statistically significant (p = 0.417). Additionally, the superior outer choroidal thickness in myopic individuals was thinner than in emmetropic individuals, but the difference was not statistically significant (p = 0.162), as shown in Table 1.

Table 1 Characteristics of retinal and choroidal structures in myopic and emmetropic subjects, all measures reported as mean ± standard deviation.

ChT = Choroidal thickness	Total	Myopic subjects	Emmetropic subjects	p-value	
Average retinal thickness (µm)	276.88 ± 12.12	274.27 ± 10.00	279.64 ± 13.54	0.002	
Average ChT (µm)	297.94 ± 57.10	287.72 ± 60.45	308.79 ± 51.41	0.009	
Centre ChT (µm)	308.17 ± 64.90	296.75 ± 67.34	320.29 ± 60.20	0.010	
Superior inner ChT (µm)	304.05 ± 62.80	300.52 ± 65.91	307.79 ± 59.43	0.417	
Superior outer ChT (µm)	309.66 ± 62.73	303.60 ± 68.19	316.09 ± 56.00	0.162	
Inferior inner ChT (µm)	317.02 ± 68.44	301.50 ± 71.40	333.50 ± 61.32	0.001	
Inferior outer ChT (µm)	318.18 ± 68.03	300.11 ± 68.01	337.39 ± 62.90	0.000	
Nasal inner ChT (µm)	291.99 ± 66.62	280.78 ± 69.46	303.91 ± 61.60	0.014	
Nasal outer ChT (µm)	250.50 ± 70.49	236.53 ± 72.53	265.34 ± 65.40	0.004	
Temporal inner ChT (µm)	299.72 ± 60.59	292.66 ± 62.59	307.23 ± 57.77	0.091	
Temporal outer ChT (µm)	282.85 ± 57.66	277.06 ± 59.79	289.06 ± 54.92	0.145	
Note:

ChT, Choroidal thickness.

Myopic subjects exhibited a decrease in mean inferior inner choroidal thickness compared to emmetropic individuals, with the difference being statistically significant (p < 0.01). Similarly, the inferior outer choroidal thickness was thinner in myopic subjects compared to emmetropic individuals, with the difference also being statistically significant (p < 0.001). Nasal inner and outer choroidal thicknesses in myopic subjects were thinner than those in emmetropic subjects, with the differences being statistically significant (p < 0.05 and p < 0.01, respectively). However, the temporal inner and outer choroidal thicknesses in myopic subjects were thinner than in emmetropic subjects, but the differences were not statistically significant (p < 0.01 and p = 0.145, respectively), as presented in Table 1.

The findings revealed a statistically significant inverse correlation (r = −0.257, p < 0.01) between mean retinal thickness and the mean spherical equivalent of myopia, as shown in Fig. 2. Similarly, there was inverse correlation (r = −0.306, p < 0.01) between mean choroidal thickness and the mean spherical equivalent of myopia, as shown in Fig. 3.

Figure 2 Scatterplot showing the correlation between mean retinal thickness and spherical equivalent of myopia.

Figure 3 Scatterplot showing the correlation between mean choroidal thickness and spherical equivalent of myopia.

Choroidal thickness across all studied areas showed weak inverse correlation with the degree of myopia, which was statistically significant (r ≤ 0.4, p < 0.05). However, the choroidal thickness in the superior outer region did not exhibit a statistically significant association (p = 0.056), as shown in Table 2.

Table 2 Correlation coefficients between choroidal thickness and degree of myopia.

	Myopic subjects n = 102 (−2.66 ± 1.59D)	
Characteristics	Mean ± S.D	Correlation co-efficient	p-value	
Centre ChT (µm)	296.75 ± 67.34	−0.300	0.002	
Superior inner ChT (µm)	300.52 ± 65.91	−0.200	0.044	
Superior outer ChT (µm)	303.60 ± 68.19	−0.190	0.056	
Inferior inner ChT (µm)	301.50 ± 71.40	−0.280	0.004	
Inferior outer ChT (µm)	300.11 ± 68.01	−0.270	0.006	
Nasal inner ChT (µm)	280.78 ± 69.46	−0.362	0.000	
Nasal outer ChT (µm)	236.53 ± 72.53	−0.377	0.000	
Temporal inner ChT (µm)	292.66 ± 62.59	−0.261	0.008	
Temporal outer ChT (µm)	277.06 ± 59.79	−0.216	0.030	
Note:

ChT, Choroidal thickness; S.D., standard deviation.

The findings revealed a non-statistically significant positive correlation (r = 0.161, p = 0.117) between mean retinal thickness and the mean spherical equivalent of emmetropia, as shown in Fig. 4. However, an inverse, non-statistically significant correlation (r = −0.196, p = 0.056) was observed between mean choroidal thickness and the mean spherical equivalent of emmetropia, as shown in Fig. 5.

Figure 4 Scatterplot showing the correlation between mean retinal thickness and spherical equivalent of emmetropia.

Figure 5 Scatterplot showing the correlation between mean choroidal thickness and spherical equivalent of emmetropia.

Discussion

Changes in retinal and choroidal structures are critical biomarkers for predicting, diagnosing, and monitoring various ocular conditions, particularly in myopia. Thus, the present study aimed to evaluate the characteristics of retinal and choroidal structures in myopic young adults using SSOCT. Our findings revealed significant insights into the morphological changes associated with myopia, contributing to a deeper understanding of its impact on ocular health. The analysis of mean retinal thickness demonstrated a marked reduction in myopic eyes compared to emmetropic eyes, with the difference being highly significant (p = 0.002). Additionally, the findings showed that retinal thickness was inversely correlated with the degree of myopia. Specifically, the retinal thickness was found to be thinner in myopes compared to emmetropes. This thinning is consistent with previous studies and suggests a correlation between increased axial length and retinal thinning (Yao et al., 2024; Cheng & Tian, 2024; Kim et al., 2020). However, a study by Aydemir et al. (2021) reported that the mean retinal layer thickness did not significantly differ between myopic and emmetropic individuals. Furthermore, the current study revealed a weak negative correlation (r = −0.257) between mean retinal thickness and the mean spherical equivalent of myopia, which was highly significant (p = 0.009). Jonas et al. (2020) suggest that retinal thinning in high myopia is primarily due to the elongation of the eye axis, which is mainly reflected in the extension of the vitreous cavity in the posterior segment of the eye. Furthermore, Wu et al. (2020) observed that in the highly myopic group, there was significant thinning of retinal and choroidal thickness, along with a decrease in retinal vessel density and retinal light sensitivity, compared to the emmetropic group. This change is likely due to the extreme expansion of the posterior segment of the eye. The mechanical stretching of the retina in myopic eyes likely contributes to these structural changes in the retina. Further studies are needed to explore the long-term implications of retinal thinning in myopic individuals, particularly concerning the risk of developing retinal pathologies.

The present study found that myopic subjects had significantly thinner mean choroidal thickness compared to emmetropic subjects, with the difference being statistically significant (p = 0.009). Additionally, choroidal thickness in all studied areas showed a strong negative correlation with the degree of myopia (p < 0.05), except in the superior outer region, where the association was not statistically significant (p = 0.056). This finding aligns with earlier reports indicating that myopia is commonly associated with choroidal thinning (Aydemir et al., 2021; Howaidy & Eldaly, 2021; Cheng & Tian, 2024). A cohort study conducted in China found that patients with myopia had thinner choroidal thickness compared to emmetropic patients, with those having high myopia showing the lowest choroidal thickness values (Wu et al., 2020). The choroid plays a crucial role in ocular blood flow and nutrient supply; therefore, reduced choroidal thickness could potentially impair these functions (Duan et al., 2019). Currently, while it is believed that the choroid has a crucial role in emmetropization, it is unclear whether it serves as an active mediator, a passive signal relay, a diffusion barrier, or a combination of these in its role to influence scleral extracellular matrix remodelling (Ostrin et al., 2023; Troilo et al., 2019). This has been confirmed in animal models of induced refractive errors (Hung, Wallman & Smith, 2000). Choroidal thinning has been observed early in myopia progression during the childhood emmetropization phase, suggesting that choroidal thickness may be an important marker for predicting myopia and its progression. Understanding the mechanisms underlying choroidal thinning in myopia would prove useful for developing targeted interventions to mitigate its effects.

Our study observed a negative correlation between the degree of myopia and both retinal and choroidal thicknesses. This relationship emphasizes the progressive nature of structural changes in the retina and choroid as myopia increases. Myopia exhibited more pronounced thinning, emphasizing the need for regular monitoring and early intervention in individuals with severe myopia to prevent potential complications. Furthermore, the findings from the present study have significant potential clinical implications. SSOCT can serve as a valuable tool for early detection and monitoring of myopia-related changes in retinal and choroidal structures. One of the significant implications of this study is the potential for using retinal and choroidal thickness measurements as biomarkers for the progression and severity of myopia. Identifying these alterations at an early stage may facilitate early intervention, potentially slowing the progression of myopia and preventing associated complications. Eye care professionals should consider including routine SSOCT assessments in the management of myopic patients, especially those with high myopia.

The present study has some limitations that warrant consideration. The sample size was relatively small, and the study population was limited to young adult males. The axial length was not measured in this study, and non-cycloplegic refraction was used. Future studies should address these limitationsincluding a larger sample size, with an equal distribution of genders, and more diverse populations to validate the current findings and explore age-related changes in retinal and choroidal structures in high myopia. Longitudinal studies are also needed to understand the progression of these structural changes over time. In spite of the abovementioned limits, our study provides valuable insights into the morphological changes associated with myopia, particularly in the retinal and choroidal layers, that could contribute to improved clinical management and intervention strategies for myopia control.

Conclusions

The study identified distinct variations in the retinal and choroidal structures between myopic and emmetropic individuals. These findings underscore the effectiveness of SSOCT in detecting morphological changes in myopic young adults. This highlights the importance of SSOCT for evaluating retinal and choroidal alterations associated with myopia. The observed thinning in these structures among myopic individuals emphasizes the need for ongoing research to understand the underlying mechanisms of myopia progression, and develop effective strategies for myopia control and management.

Supplemental Information

Supplemental Information 1 Raw data.

Supplemental Information 2 Strobe statement.

Additional Information and Declarations

Competing Interests

The authors declare that they have no competing interests.

Author Contributions

Saif Hassan Alrasheed conceived and designed the experiments, performed the experiments, analyzed the data, prepared figures and/or tables, authored or reviewed drafts of the article, and approved the final draft.

Yazan Gammoh conceived and designed the experiments, performed the experiments, authored or reviewed drafts of the article, and approved the final draft.

Human Ethics

The following information was supplied relating to ethical approvals (i.e., approving body and any reference numbers):

Ethical permission was obtained from the Qassim University Health Research Ethics Committee (approval number 21-10-03), and the study adhered to the Declaration of Helsinki guidelines. Written informed consent was obtained from all participants after explaining the study’s aim. Participants were able to withdraw from the study at any time, and no remuneration was offered for participation. Data were collected confidentially without any individual information being recorded.

Data Availability

The following information was supplied regarding data availability:

The raw measurements are available in the Supplemental File.

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
