# Peer review of "Characteristics of the retinal and choroidal thicknesses in myopic young adult males using swept-source optical coherence tomography"

_PeerJ, doi:10.7717/peerj.19030_

## Round 0.1 · original submission · Major Revisions

Please pay particular attention to the comments from Reviewer 1, who has recommended rejection overall.

Two of the reviewers have suggested citations in parts of their review. You do not have to include these citations unless you feel they are relevant.

Reviewer 1 ·

Basic reporting

The authors have presented, analyzed and discussed data on the differences in macular retinal and choroidal thickness in myopic young adults compared to age-matched emmetropes using swept-source optical coherence tomography. Further, they explored associations between refractive error and retinal/choroidal thickness in myopes. While the data in this study and its relevance are very useful, there are significant and minor issues with the paper which are detailed below.

There are very few typographical and grammatical errors in the manuscript.

There are redundancies in the report of data.

(See attached document/'Additional comments' for more details).

Experimental design

There are significant issues with the study design such as the absence of axial length data, selection of only males, and the process of analysis of OCT images.
These fundamental issues are not discussed.

(See attached document/'Additional comments' for more details).

Validity of the findings

When reviewing the raw data from the attached dataset “Raw_Data_Final.xlsx”, I noticed my analyses were different from that submitted by the authors. After a simple count of the number of myopes and emmetropes from the data, there is a discrepancy in the distribution. The dataset shows 101 myopes and 97 emmetropes compared to the author’s report of 102 myopes and 96 emmetropes. This among other reasons may explain the disagreement between the results of my analysis and the authors’.

This warrants a complete reanalysis of the data.

(See attached document/'Additional comments' for more details).

Additional comments

The authors have presented, analyzed and discussed data on the differences in macular retinal and choroidal thickness in myopic young adults compared to age-matched emmetropes using swept-source optical coherence tomography. Further, they explored associations between refractive error and retinal/choroidal thickness in myopes. While the data in this study and its relevance are very useful, there are significant (**) and minor issues with the paper which are detailed below:

Title:
L1-2: Since thickness was the only characteristic of the retina and choroid analyzed and discussed in the paper, I suggest replacing ‘structures’ with ‘thicknesses’ to reflect that.
Also, since only males were included in this study, this should be reflected in the title also.

Abstract:
L26 & L33: As this was a cross-sectional study and from the results, the authors did not assess ‘changes’. Please rephrase the related sentences to reflect that.
L28-25: Please include in the methods of the abstract that only male subjects were recruited for this study.
L29-31: The authors assessed only thicknesses in this study. No mention of any other measure of retinal or choroidal ‘structural integrity’ was evaluated. Please remove ‘structural integrity’.
L34: To maintain consistency in the report of variables, please report the refractive error data similar to that of age. Thus, only state the unit after the standard deviation: -2.66 ± 1.59D. This should be corrected in all aspects of the manuscript.
L34-41: To maintain consistency in the report of p-values to facilitate easy reading of the manuscript, please report the ‘p’ as a lower case. As a suggestion, the authors can choose to report significant values as p<0.05, p<0.01 and p<0.001. Non-significant values can be reported with the true values. This should be corrected in all aspects of the manuscript.
L34-35: Please report the refractive data for the emmetropic group since that of the myopic group was reported.

Introduction:
**L49-85: While the focus of this study is to describe the ‘Characteristics of the retinal and choroidal structures …’, only the choroid is described in the entire section with the retina largely ignored. Please address this.
L49-64: The first paragraph does not convey a common theme. Part of the paragraph describes the choroid while other parts describe optical coherence tomography. Please restructure the paragraph.
L50: The authors point to the choroid as forming the ‘retinal barrier’. Please explain this as I am unfamiliar with the term. If it was referring to the blood-retina barrier, specifically the outer blood-retinal barrier, then that is incorrect as that barrier is formed by the tight junctions of the retinal pigment epithelium (of the retina) to control movement of solutes and nutrients from the choroid to the sub-retinal space (Naylor et al., 2019; Cunha-Vaz et al., 2011). The authors can provide references to dispute that or correct that assertion.
• Naylor A, Hopkins A, Hudson N, Campbell M. Tight Junctions of the Outer Blood Retina Barrier. Int J Mol Sci. 2019;21(1):211.
• Cunha-Vaz J, Bernardes R, Lobo C. Blood-retinal barrier. Eur J Ophthalmol. 2011;21 Suppl 6:S3-S9.
L68: The authors point to ‘High myopia’ as the risk factor to developing sight threatening conditions. While that is true, it has been shown that any degree of myopia is associated with these blinding complications with the risk increasing with increasing degrees of myopia: shown in Figures 1-4 of Reference #11. Further, since the average myopia in this study is on the low side and no sub analysis on high myopic subjects, it is best to change ‘High myopia’ to ‘Myopia’ to better suit this study.
L80-82: Please provide a citation to support the assertion ‘as this demographic is at a critical stage where myopia progression is often most pronounced.’

Methods:
There are significant issues with the study design such as the absence of axial length data, selection of only males and analysis of OCT images.
**L88-90: Please include at this point that ONLY males were included for this study. Also provide a justification for exclusion of females in this study.
L93: In the methods, the authors state that the participants recruited were from age 18-30 years. However, in the Results (L131), the authors changed this to 18-25 years. Please which is the correct age range.
L98: Please provide the mathematical definition of SER.
L98-99: Please provide a citation for the classification of myopia and emmetropia into these ranges. Please indicate a methodologic and statistical reason for this choice of classification.
Further, why was the limit of myopia capped at -7.00D. Were subjects with myopia above -7.00D excluded from the study? If so, provide a justification for that.
L100: The authors list ocular pathology as an exclusion criterion. Were patients with ocular pathologies such as mild allergic conjunctivitis, mild dry eyes, pterygium (grade 1), or pinguecula that have little to no effect on retinal imaging excluded from the study? If not, then to avoid ambiguity, I suggest the authors list the specific ocular pathologies or use a narrower term.
**However, data on axial length is missing from this study. Absence of axial length measurement raises fundamental issues with this study.
First, from extensive reports, magnification correction is very necessary for accurate assessment of retinal and choroidal thickness in myopes, and especially important in select similar retinal and choroidal regions for comparison (Deng et al., 2022; Nowroozizadeh et al., 2014; Hirasawa et al., 2014). Axial length is the metric used for magnification correction; for either Kang's and Littmann's/Bennett’s methods (Hirasawa et al., 2014b).
• Deng J, Jin J, Zhang B, et al. Effect of Ocular Magnification on Macular Choroidal Thickness Measurements Made Using Optical Coherence Tomography in Children. Curr Eye Res. 2022;47(11):1538-1546.
• Nowroozizadeh S, Cirineo N, Amini N, et al. Influence of correction of ocular magnification on spectral-domain OCT retinal nerve fiber layer measurement variability and performance. Invest Ophthalmol Vis Sci. 2014;55(6):3439-3446.
• Hirasawa K, Shoji N, Yoshii Y, Haraguchi S. Determination of axial length requiring adjustment of measured circumpapillary retinal nerve fiber layer thickness for ocular magnification. PLoS One. 2014a;9(9):e107553.
• Hirasawa K, Shoji N, Yoshii Y, Haraguchi S. Comparison of Kang's and Littmann's methods of correction for ocular magnification in circumpapillary retinal nerve fiber layer measurement. Invest Ophthalmol Vis Sci. 2014b;55(12):8353-8358.
Further, it is widely accepted that the driver of retinal and choroidal changes, is the axial length elongation (pointed out by authors in L192-194) and not the refractive error (which is the consequence of axial length changes) per se, as individuals with refractive myopia are not expected to have these posterior segment changes. Thus, without providing axial length measures and showing the myopia within this population is driven by axial length changes, deductions drawn from this study are shaky.
L111-113: Please provide the details of the instruments used: the autorefractor and SS OCT. These details should include the version of the instrument, manufacturer and location.
From the attached dataset, visual acuity was measured. Please include details of the measurement of visual acuity in the methods such as the chart used and (if possible, details of the manufacturer and location).
**L97-99 & L11-112: Was refraction measured after cycloplegia? Current standards highly recommend cycloplegia as the “gold standard” for any refraction-related study (Flitcroft et al., 2019). The absence of cycloplegic refraction data despite the mention of ‘complete medical records’ presents severe limitations to the study. If (cycloplegic) refraction data is available, this should be included in the study.
• Flitcroft DI, He M, Jonas JB, et al. IMI - Defining and Classifying Myopia: A Proposed Set of Standards for Clinical and Epidemiologic Studies. Invest Ophthalmol Vis Sci. 2019;60(3):M20-M30.
L113: Were the ‘structural integrity’ of the retina and choroid assessed qualitatively or quantitatively? As no such metric was provided in the Results section.
L113: While the authors describe measurement of both the retinal and choroidal thickness at nine predefined locations, the retina was only reported as mean of the quantified area while the choroid was reported for different locations. Please provide a reason for this.
**L113-114: Statistical measures of accuracy and repeatability of the measurement methods used by the tomographer needs to be provided.
**L114-118: Please provide extensive details of the scan parameters including but not limited to the dimensions (area) of the scan as well the number of A-lines per B-scan and number of B-scans per image etc.
Also, provide a cutoff of image quality (such as Signal Strength Index) for selection of images for further analysis.
**L113-118: Were the OCT scans manually or automatically segmented. If automated, were segmentation errors checked for and corrected? Please provide details.
Further, please define the boundaries of the retina and choroidal thickness using anatomical landmarks.
L117: Define the size of the sub-fovea.
L115-118: The description of the measurement appears as if the thicknesses were quantified over the Early Treatment Diabetic Retinopathy Study (ETDRS) map. If so, mentioning that will be helpful for readers to visualize the quantification as the map/grid is commonly used in OCT images.
Providing a Figure which shows a sample OCT scan as well as an overlay of the ETDRS map (on the en-face image) and segmentation lines (on the B-scan) is highly recommended to enhance comprehension of the methods.
L120-121: Please provide the details of the version of Microsoft Excel used. Further provide details of the developer/manufacturer and location of both statistical software packages used.
L122-123: Since all the data was normally distributed and median was never used in describing the data, please remove it.
L123-125: While technically SPSS ANOVA analysis of 2 groups will yield the similar results as an independent t-test, the independent t-test is more appropriate in this study since only 2 groups (myopes vs emmetropes) compared. ANOVA is more suitable for comparing 3 or more groups.
L125-126: Please provide a justification for why correlation analysis was focused on only myopes.
L126: Please replace ‘structural changes’ with ‘thickness’ as that is more appropriate as explained before.
L126: Please indicate what specific correlation was used. For example, Pearson or Spearman.

Results:
**L130-174: When reviewing the raw data from the attached dataset “Raw_Data_Final.xlsx”, I noticed my analyses were different from that submitted by the authors. After a simple count of the number of myopes and emmetropes from the data, there is a discrepancy in the distribution. The dataset shows 101 myopes and 97 emmetropes compared to the author’s report of 102 myopes and 96 emmetropes. This among other reasons may explain the disagreement between the results of my analysis and the authors’. Please comment on this.

L138-140: Just reporting that the K-S test showed variables were normally distributed with p>0.05 should suffice. There is no need for Table 1. The keys of a & b below Table 1 are not reflected in the Table itself.
Also, if it was not performed, the authors should the normality of age.
**L142-160: The authors basically describe every aspect of Table 2 and in details and did not reference it in text. This therefore makes Table 2 redundant. It will be best to summarize the key findings in Table 2 in a couple of sentences and reference the table in the text.
L161-163: The sentence needs to be reworked. The authors point to ‘statistically significant’ and then further state ‘was highly significant’. This is repetition. Please remove the phrase ‘which was highly significant (P = 0.009)’, remove ‘**’ and add the p-value to the correlation coefficient (r).
L166: A correlation co-efficient of -0.306 is not ‘strong’. It can be interpreted as ‘weak’. Also, remove ‘**’ from the ‘r=-0.306’.
L161-165: Please reference the appropriate Table or Figure in text next to the appropriate description.
L166-169: None of the correlations in Table 3 are strong (r ≥ 0.6). They are all ‘weak’ (r ≤ 0.4). Please correct this.
L170-174: Please remove L170-174 as it is a repetition of L161-165. If they are Figure captions, there are already captions on both Figures 1 & 2.

Table 2: Please address the following issues:
i. Keep all values to 2 decimal places.
ii. Indicate that all measures all reported as mean ± standard deviation.
iii. Add ‘subjects’ to ‘Myopic’ and ‘Emmetropic’ in row 1.
iv. Change “Choroidal Thickness= ChT” to “ChT = Choroidal Thickness”.

Table 3: Please address the following issues:
i. Keep all values to 2 decimal places.
ii. Change ‘Myopia’ in row 1 to ‘Myopic subjects’.
iii. Change ‘Correlation’ in row 2 to ‘Correlation co-efficient’.
iv. Add “ChT = Choroidal Thickness” and “S.D.= Standard deviation” to bottom of Table.

Table 3, Figures 1 & 2: There is redundancies in the report of information. The 3rd and 4th row of Table 3 are basically the data plotted for Figure 1 and 2 respectively. The authors need to either remove row 3 and 4 of Table 3 and keep Figures 1 & 2 or keep Table 3 as it is and remove Figure 1 and 2.

If keeping Figures 1 & 2, address the following issues:
i. Please add tick marks (preferred) or gridlines to enhance reading of the Figures.
ii. Also, remove equation from the line. Remove it entirely or add it to the coefficient of determination (R2) on the top left side of the image as it is currently obscuring data points.
iii. Just have ‘R2’ and not ‘R2 Linear’ as it is evident the line of best fit is linear.


This study could benefit from an analysis reporting the topographical thickness variation and asymmetry among the nine predetermined locations (for retina and choroid) in emmetropic subjects.
Further, a model (possibly a stepwise multiple regression model) to determine which retinal and choroidal region was most correlated with refraction (and/or axial length) will be informative.




Discussion:
L189-190: The authors should compare similar studies (in terms of methods) to theirs and discuss. The cited literature explored peripapillary retinal nerve fiber layer thickness, which the authors wrongly described as retinal thickness. This measure is also fundamentally different from the macular thickness in this study. Please rework this aspect of the discussion.
L191: A correlation co-efficient of “-0.257” is ‘weak’ not ‘strong’. Please correct this.
L196-197: Can the authors clarify what is ‘middle blood vessels’? I am unfamiliar with this blood vessel in the eye.

L210: The authors assert that the “Choroidal tissue supports axial elongation by reshaping the scleral extracellular matrix …”. The cited literature (Ref # 25) does not support this assertion. Please provide an appropriate reference.
Currently, while it is believed that the choroid has a crucial role in emmetropization, it is unclear whether it serves as an active mediator, a passive signal relay, a diffusion barrier, or a combination of these in its role to influence scleral extracellular matrix remodeling (Ostrin et al., 2023; Troilo et al., 2019).
• Ostrin LA, Harb E, Nickla DL, et al. IMI-The Dynamic Choroid: New Insights, Challenges, and Potential Significance for Human Myopia. Invest Ophthalmol Vis Sci. 2023;64(6):4.
• Troilo D, Smith EL 3rd, Nickla DL, et al. IMI - Report on Experimental Models of Emmetropization and Myopia. Invest Ophthalmol Vis Sci. 2019;60(3):M31-M88.
L219: The statement ‘High myopia exhibited more pronounced thinning’ is not supported by any analysis in this study. In order to make this claim, the authors need to include a sub-group analysis of high myopes in their results.
**L230: The authors state that a limitation of their study is a relatively small sample size. Did the authors perform an a priori sample size analysis to determine if their sample size falls short of the required number needed to make statistical inferences about the particular population at a specific effect size or power? Please comment.


**There are oddities in the consent form attached. While the objective in the consent form [“To investigate structural changes in the retina and choroid in myopic young adults using Swept-Source Optical Coherence Tomography (SSOCT)”] is consistent with this study, the procedure (“In one visit, your accommodation functions will be check”) and expected benefits (“This study could lead to identifying the normative
data for accommodation function for Saudi young adults in the Qassim region.”) are not in line with the current study. Please explain.

Annotated reviews are not available for download in order to protect the identity of reviewers who chose to remain anonymous.

·

Basic reporting

1-Please clarify certificate approval number
2- Please clarify artificial intelligence statement disclosure

Experimental design

1- Please clarify inclusion criteria in details
2- Please clarify methods of examinations of visual acuity, anterior segment, IOP, posterior segment
3- More details regarding SS-OCT technique

Validity of the findings

OOD

Additional comments

1- Conclusion should be condensed
2- Please added recent references

·

Basic reporting

The thesis is well written and I find no major flaws in it. The only minor point, which is also pointed out by the authors, is the representation of only the male population.

Experimental design

No comments

Validity of the findings

No comments

Reviewer 4 ·

Basic reporting

The authors have studied the retinal and choroidal thickness profiles in eyes with different refractive errors.
Please mention in the introduction that myopic eyes have greater RNFL progression than non-myopic eyes (Cite: DOI: 10.1097/OPX.0000000000001519), and existing OCT flags myopic eyes as falsely negative abnormal RNFL (Ref: doi: 10.101/jamaophthalmol.2016.2343).

Experimental design

3. How were the retinal and choroidal thicknesses measured manually/automatically? Please give details of the scan taken (degrees).

Validity of the findings

Axial length was not measured. This is a limitation, as retinal and choroidal thickness changes are strongly related to axial length and less with refractive error. Please mention this in the discussion.

Additional comments

none

---

## Round 0.2 · accepted · Accept

I have now had the opportunity to read your revised manuscript, and your responses to the reviewers' comments. I believe that you have addressed the concerns raised, and I am happy to accept your manuscript.

Reviewer 1 ·

Basic reporting

See attached document

Experimental design

See attached document

Validity of the findings

See attached document

Additional comments

See attached document

Annotated reviews are not available for download in order to protect the identity of reviewers who chose to remain anonymous.

·

Basic reporting

Good

Experimental design

Good

Validity of the findings

Excellent

Additional comments

non

·

Basic reporting

No comment

Experimental design

No comment

Validity of the findings

No comment

Reviewer 4 ·

Basic reporting

None

Experimental design

None

Validity of the findings

None

Additional comments

None